# Pulmonary mucoid *Pseudomonas aeruginosa* infection and association with higher species richness and stronger inflammatory immune response

Chenyan Zhang,[1] Zhengke Sun,[2] Yuanhua Lin,[1] Wenfang Long,[1] Rongguang Zhang,[1] Hairong Huang,[1] Wenjuan Liang[1]

**ABSTRACT** The mucoid phenotype of *Pseudomonas aeruginosa* (PA) is regarded as a comprehensive adaptive stress response to difficult environmental circumstances. However, there is little knowledge about the relationship between the prevalence of mucoid PA and species richness and immune inflammatory response. A case-control study was conducted in hospitalized patients with pulmonary infections caused by mucoid and non-mucoid PA. Sputum samples were subjected to 16S rDNA sequencing to characterize microbial diversity and taxonomic composition, while serum levels of TNF-α, IL-6, IL-8, IL-10, and IL-17 were measured using enzyme-linked immunosorbent assays. Subsequent statistical analysis using R 4.0 revealed significant correlations between differentially abundant microbial taxa and cytokine profiles. Compared to the non-mucoid PA group, the mucoid PA group demonstrated significantly higher α-diversity indices in terms of species richness, as indicated by the Chao1 ($P = 0.0015$) and Observed-species metrics ($P = 0.0014$). Furthermore, distinct β-diversity patterns were observed between the two groups ($P < 0.05$). LefSe analysis revealed significant enrichment of *Veillonella* spp., *Haemophilus* spp., *Porphyromonas* spp., *Prevotella* spp., *Actinomyces* spp., *Lactobacillus* spp., and *Rothia* spp. in the mucoid PA group, while *Stenotrophomonas* spp., *Acinetobacter* spp., *Parvimonas* spp., and *Serratia* spp. dominated in the non-mucoid PA group. The mucoid PA infections showed marked elevation of IL-8 ($P = 0.0137$), TNF-α ($P = 0.0048$), IL-10 ($P = 0.0042$), IL-17 ($P = 0.0220$), and IL-6 ($P = 0.0001$). Spearman correlation revealed *Veillonella* spp./*Rothia* spp./*Porphyromonas* spp./*Prevotella* spp. positively correlated with IL-10/TNF-α/IL-17/IL-6, whereas *Haemophilus* spp. showed a negative relationship with IL-17. *Stenotrophomonas* spp. exhibited strong negative correlations with IL-10/IL-6, and *Serratia* spp. was inversely associated with TNF-α in non-mucoid PA infections. Clinically distinct microbial ecosystems in mucoid PA correlate with exacerbated inflammation. This phenotype-driven dichotomy provides actionable biomarkers for stratified antimicrobial/immunomodulatory therapies in chronic lung disease.

**IMPORTANCE** This study holds significant clinical and scientific importance, as it elucidates the critical differences between mucoid and non-mucoid *Pseudomonas aeruginosa* (PA) infections in pulmonary patients. By demonstrating that mucoid PA infections are associated with distinct microbial ecosystems (higher species richness and different taxonomic compositions) and more severe inflammatory responses (elevated TNF-α, IL-6, IL-8, IL-10, and IL-17), the research provides crucial insights into phenotype-specific pathogenesis. The identified correlations between specific bacterial species (e.g., *Veillonella*/*Rothia* with pro-inflammatory cytokines) offer potential biomarkers for clinical stratification. These findings are particularly valuable for developing targeted therapeutic strategies, as they suggest mucoid PA infections may require different antimicrobial/immunomodulatory approaches compared to non-mucoid variants. The

**Peer Reviewers** Shuaiyin Chen, Zhengzhou University, Zhengzhou, China; Yongbin Wang, Xinxiang Medical University, Xinxiang, China

Address correspondence to Wenjuan Liang, wenwen3_1@126.com.

Chenyan Zhang and Zhengke Sun contributed equally to this article. Author order was determined by drawing straws.

Hairong Huang and Wenjuan Liang contributed equally to this article.

The authors declare no conflict of interest.

See the funding table on p. 10.

study bridges an important knowledge gap in understanding how bacterial phenotypic adaptation influences host-microbiome interactions and disease outcomes in chronic lung infections.

KEYWORDS    mucoid *Pseudomonas aeruginosa*, sputum microbiome, immune factors

Lower respiratory tract infections—encompassing pneumonia and suppurative pulmonary complications—constitute the fourth-largest global mortality driver according to GBD 2021 data (1, 2). In China's nosocomial infection landscape, *Pseudomonas aeruginosa* (PA) emerges as the predominant gram-negative pulmonary pathogen, demonstrating a triple threat of high prevalence, extensive antimicrobial resistance, and rapid genomic adaptability (3, 4). Particularly in patients with underlying pulmonary comorbidities (chronic obstructive pulmonary disease, bronchiectasis), these evolutionary advantages facilitate persistent airway colonization, driving therapeutic failure rates exceeding 58% in CF populations (5).

The mucoid phenotypic variant (mucoid PA, MPA) activates alginate biofilm synthesis via the alg operon, exhibiting superior immune evasion and β-lactam resistance compared to non-mucoid PA (NMPA), thereby amplifying risks of hospital-acquired device-related infections (6–8). Metagenomic studies confirm that dysbiosis in the lower respiratory microbiota correlates closely with disease progression, where pneumonia markedly reduces microbial diversity and promotes pathogen proliferation (9–17). Current PA research predominantly focuses on single phenotypes, lacking parallel comparisons between MPA and NMPA, particularly regarding the microbiota-immune interaction mechanisms underlying mucoid variant infections (18, 19). The prevailing focus on single-phenotype PA models has obscured critical differences in pathogenesis and therapeutic susceptibility between MPA and NMPA. Clinically, this gap contributes to undifferentiated treatment protocols and persistent infections in chronic respiratory diseases.

This study systematically investigates sputum microbiota disparities between MPA- and NMPA-infected patients via 16S rDNA sequencing and immune factor profiling, aiming to elucidate correlations between differential microbial abundance and serum immune indices. By exploring microbiota-immune regulatory networks, this work seeks to advance novel therapeutic strategies for MPA-associated infections.

## MATERIALS AND METHODS

### Study population

Hospitalized patients with pneumonia, confirmed by positive *P. aeruginosa* isolation in two consecutive cultures and characteristic imaging findings, admitted from August 2023 to January 2024 at a tertiary hospital in Haikou city, were enrolled, according to Guidelines for the Diagnosis and Treatment of Community-Acquired Pneumonia in Chinese Adults (2016 Edition). Exclusion criteria included additional malignancies, mixed infections, incomplete clinical records, and serious comorbidities. This study employed a case-control design, in which participants were classified into the MPA group and the NMPA group. No statistically significant differences were observed in age, sex, or clinical characteristics between the two groups, indicating adequate comparability.

### Sputum sample collection, culture, and identification

Deep sputum samples from the lower respiratory tract, collected after oral decontamination, were obtained in the morning and cultured on MacConkey agar at 35°C for 24 hours to identify dominant pathogens.

## Baseline data and hematologic parameter collection

Demographic data and hematologic parameters—including white blood cell count, lymphocyte percentage/absolute count, neutrophil percentage/absolute count, and high-sensitivity C-reactive protein—were extracted from electronic medical records.

## Immune factor quantification

Fasting venous blood samples collected on the second day of hospitalization were analyzed using enzyme-linked immunosorbent assays (ELISA) kits (Fankewei Biotech) to quantify TNF-α, IL-6, IL-8, IL-10, and IL-17 levels. Absorbance was measured with a Thermo Fisher microplate reader, and concentrations were calculated using standard curves.

## Genomic DNA extraction, purification, amplification, and sequencing

Sputum DNA was extracted and purified using commercial kits, quantified via Nanodrop, and amplified using V3-V4 primers (forward: ACTCCTACGGGAGGCAGCA; reverse: GGA CTACHVGGGTWTCTAAT). PCR products were sequenced on the Illumina HiSeq 2500 platform by Personalbio (Shanghai, China; Project ID: MD202402291325EZU2; Contract: PN20240131006; Proposal: MbPL2024021049).

## Microbiome analysis

Alpha-diversity indices (Chao1, Simpson, Shannon, and Observed species) were calculated using QIIME2, with sequencing saturation validated via rarefaction curves. ASV-based Venn diagrams visualized shared/unique taxa between groups. β-diversity analysis employed UniFrac distance matrices, visualized through principal coordinate analysis (PCoA). PERMANOVA tested structural differences between cohorts. LEfSe identified differentially abundant taxa (LDA > 2.0, $P < 0.05$), and bar charts illustrated compositional profiles.

## Statistical analysis

SPSS 26.0 processed data: categorical variables were expressed as frequencies (%), normally distributed continuous variables as mean ± SD, and non-normal data as median (P25, P75). Group comparisons used $t$-tests (normal data) or Mann-Whitney U tests (non-normal data), with $\chi^2$ tests for categorical associations. GraphPad Prism 9.5 facilitated visualization and significance annotation. R 4.0 performed Spearman correlation analyses to generate heatmaps of differential taxa and immune factors. Statistical significance was set at $P < 0.05$.

## RESULTS

### Demographic and clinical characteristics of patients

A total of 60 patients were enrolled in the study, comprising 40 in the NMPA group and 20 in the MPA group. As detailed in Table S1 (Supplementary Material), no statistically significant differences were observed between the two groups in baseline characteristics, including demographics (sex, female: 70.0% vs 50.3%; age: 65.55 ± 14.17 vs 66.78 ± 14.81 years; smoking; alcohol use), comorbidities (hypertension, diabetes mellitus, cerebrovascular disease, or cardiac disorders), or hematologic parameters ($P > 0.05$ for all group comparisons).

### Basic 16S rDNA sequencing metrics

All 60 samples yielded sequences ranging from 230 to 443 bp, with an average length of 425 bp (predominantly 401–433 bp; Fig. S1). Operational Taxonomic Unit (OTU) classification identified 1,558 families, 6,125 genera, and 2,513 species. Taxonomic

annotation predominantly resolved microbial composition at family and genus levels. OTU distributions across classification tiers are detailed in Fig. S2.

## Comparative analysis of microbial α- and β-diversity between groups

Alpha diversity was calculated by the species richness indices (Chao) and species diversity indices (Shannon). Compared to the non-mucoid PA group, the mucoid PA group demonstrated significantly higher α-diversity indices in terms of species richness, as indicated by the Chao1 (284.64 ± 158.70 vs 175.16 ± 143.85, $P = 0.0015$) and Observed-species metrics (265.64 ± 149.81 vs 164.51 ± 140.93, $P = 0.0014$), though no significant difference in the Shannon diversity index between the two groups ($P = 0.41$; Fig. 1A).

Principal coordinate analysis (PCoA) revealed that PCo1 and PCo2 accounted for 4.7% and 2.5% of total variance, respectively. PERMANOVA confirmed significant microbial community divergence between NMPA and MPA groups ($P < 0.05$; Fig. 1B).

## Microbial composition analysis between MPA and NMPA groups

As depicted in Fig. 2, the phylum-level composition of sputum microbiota in both groups was dominated by *Proteobacteria* (56.65% in MPA vs 52.51% in NMPA). Subsequent dominant phyla included *Firmicutes* (17.25% in MPA vs 13.67% in NMPA), *Bacteroidetes* (11.99% in MPA vs 9.43% NMPA), and *Actinobacteria* (9.94% in MPA vs 20.83% in NMPA). At the genus level (Fig. 3), the top three taxa in the NMPA group were *Corynebacterium* (17.42%), *Stenotrophomonas* (6.92%), and *Streptococcus* (6.15%), whereas the MPA group exhibited dominance of *Veillonella* (6.31%), *Streptococcus* (5.88%), and *Corynebacterium* (4.43%). Notably, the abundance of *Actinobacteria* (9.94% in MPA vs 20.83% in NMPA, $P < 0.05$) and *Corynebacterium* (4.43% vs 17.42%; $P < 0.01$) was significantly elevated in the NMPA cohort.

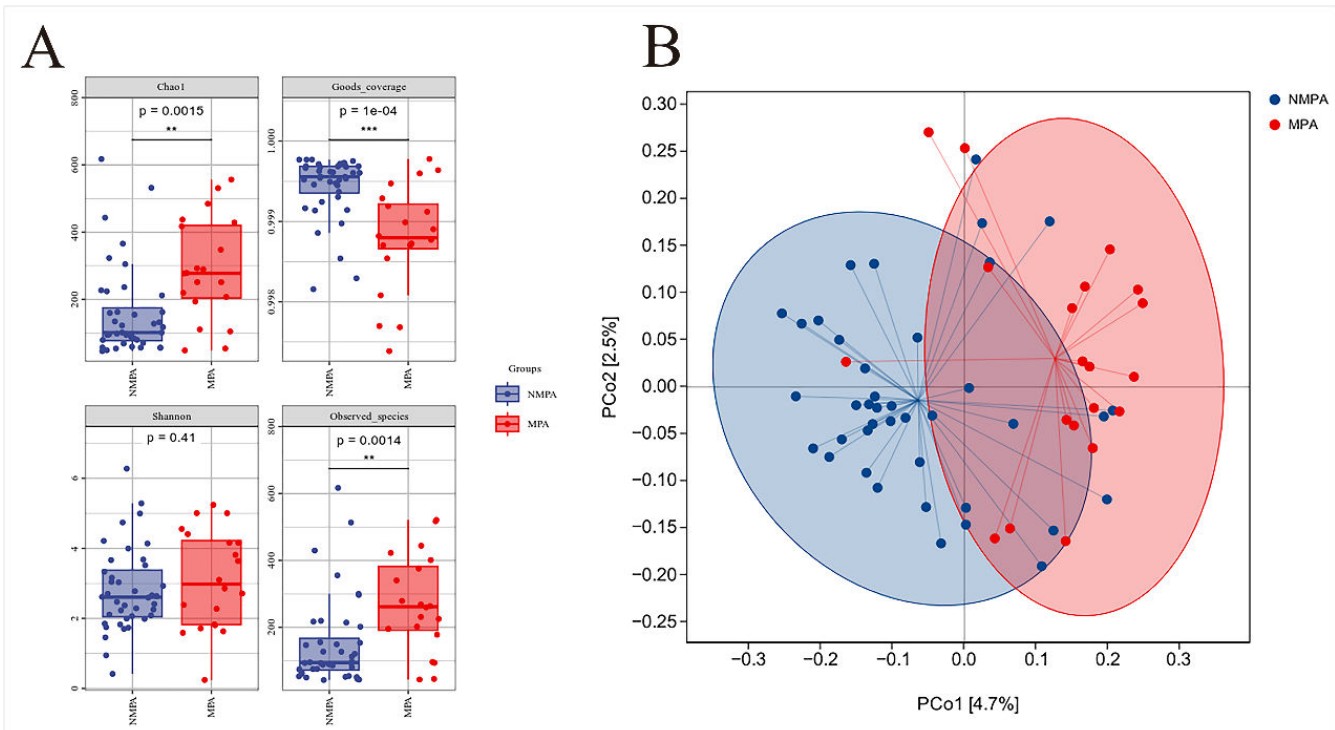

**FIG 1** α- and β-diversity of the sputum microbiota in the MPA and NMPA group. Panal A displays the distribution differences of Chao1, Good's coverage, Shannon, and Observed-species across the MPA and the NMPA group using boxplots. Panel B shows a distribution pattern of partial overlap but an overall separation trend, indicating differences in microbial community structure between the two groups.

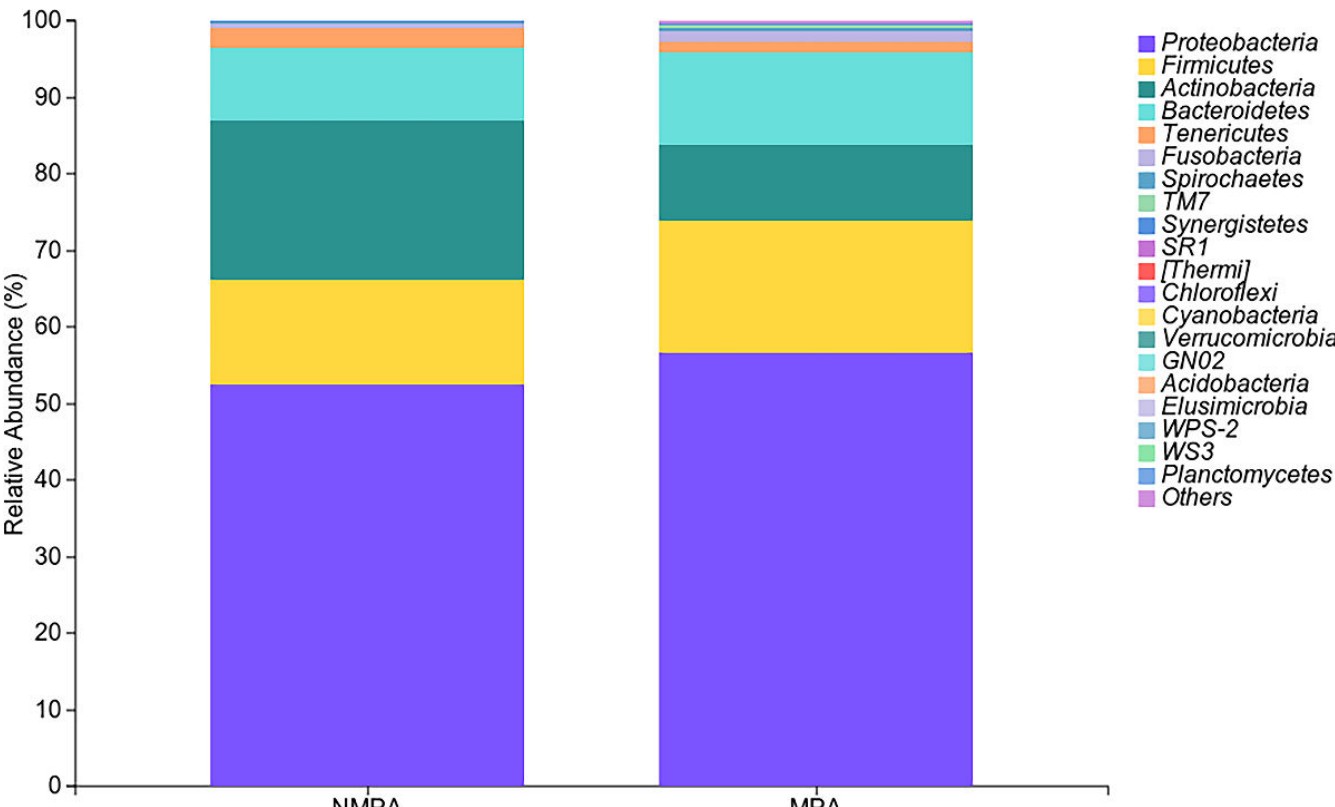

**FIG 2** Composition of the microbiota at the genus level in the NMPA and MPA groups. The stacked bar chart visually displays the relative abundance (%) of bacterial phyla in the MPA and the NMPA group, and the legend on the right clearly labels the colors corresponding to each bacterial phylum, as *Proteobacteria* is the dominant group, and the second most abundant phylum is *Firmicutes*.

## Differential microbial abundance between cohorts

Linear discriminant analysis effect size (LEfSe) identified 50 significantly divergent taxonomic units (LDA > 2.0, $P < 0.05$). Sixteen taxa were enriched in the NMPA group, while 34 exhibited MPA-specific dominance (Fig. 4). The MPA group showed significant enrichment of *Veillonella*, *Haemophilus*, *Porphyromonas*, *Prevotella*, *Actinomyces*, *Lactobacillus*, and *Rothia*. In contrast, the NMPA group demonstrated higher abundance of *Stenotrophomonas*, *Acinetobacter*, *Parvimonas*, and *Serratia*.

## Immune factor profiles between groups

ELISA revealed significantly elevated serum levels of IL-8 (1.20 vs 0.94, $P = 0.0137$), TNF-α (0.52 vs 0.50, $P = 0.0048$), IL-10 (0.69 vs 0.65, $P = 0.0042$), IL-17 (0.64 vs 0.61, $P = 0.0220$), and IL-6 (1.20 vs 0.94, $P = 0.0001$) in the MPA group compared to NMPA (Fig. 5).

## Correlation between differential taxa abundance and immune factors

Spearman correlation analysis (Fig. 6) was performed on the top 10 genera (by relative abundance) and immune markers (IL-10, IL-8, TNF-α, IL-6, and IL-17). In the MPA group, *Veillonella*, *Rothia*, *Porphyromonas*, and *Prevotella* exhibited positive correlations with IL-10, TNF-α, IL-17, and IL-6 ($r > 0.4$, $P<0.05$), while *Haemophilus* showed a negative correlation with IL-17 ($r = -0.38$, $P = 0.02$). In the NMPA group, *Stenotrophomonas* displayed strong negative correlations with IL-10 and IL-6 ($r < -0.45$, $P < 0.01$), and *Serratia* was inversely associated with TNF-α ($r = -0.32$, $P = 0.04$).

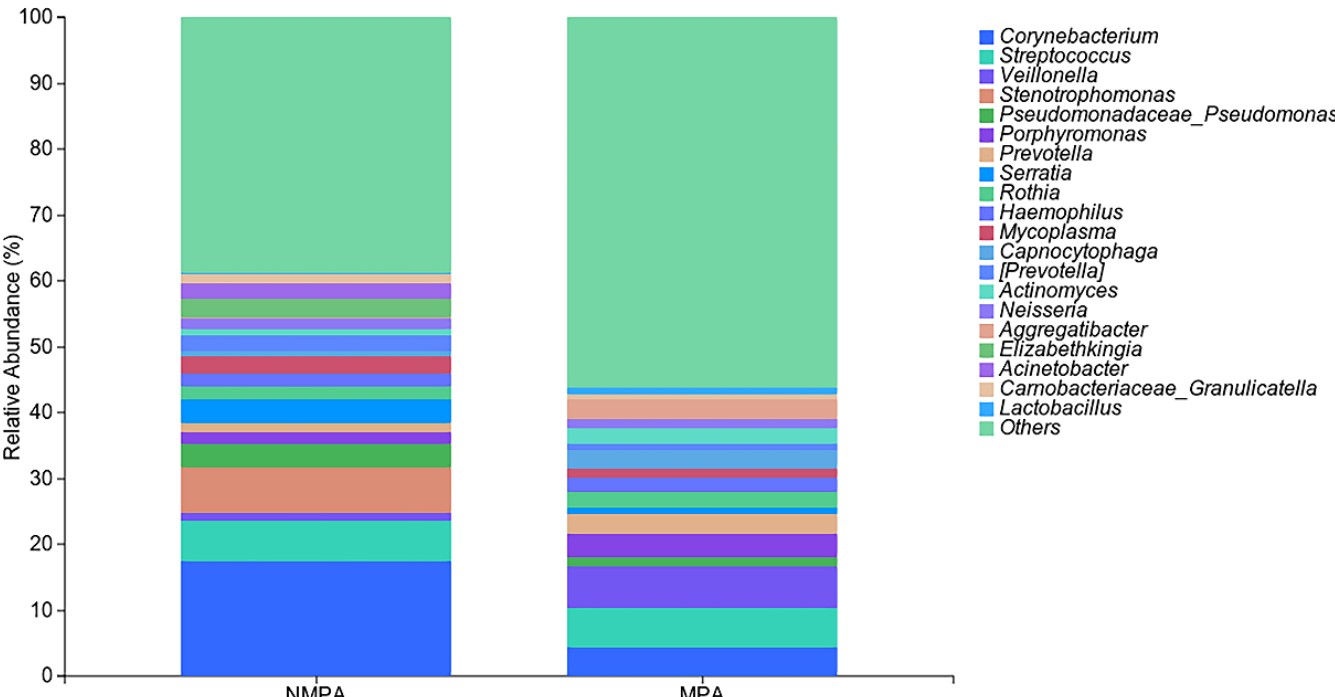

**FIG 3** Composition of the microbiota at the species level in the NMPA and MPA groups. The relative abundance (%) at the genus level in the MPA and NMPA groups is displayed, and the top taxon in the NMPA group is *Corynebacterium,* whereas the MPA group exhibits dominance of *Veillonella*.

## DISCUSSION

This study revealed that sputum microbiota in patients with MPA pulmonary infections exhibited higher α-diversity (species richness) compared to those with NMPA infections, suggesting that MPA-infected airways develop distinct microbial community structures characterized by increased species richness without substantial changes in community evenness. At the phylum level, both groups were dominated by *Proteobacteria*, *Firmicutes*, *Bacteroidetes*, *Actinobacteria*, and *Tenericutes*. Notably, genus-level analysis identified significant enrichment and elevated abundance of *Veillonella*, *Rothia*, *Haemophilus*, *Porphyromonas*, and *Prevotella* in MPA patients.

*Veillonella*—a gram-negative anaerobic bacterium (20)—has been consistently associated with increased prevalence across a range of pathological conditions. Notably, elevated abundance of *Veillonellaceae* has been reported in clinical and preclinical studies evaluating Shenling Baizhu San, a traditional herbal formulation, for the treatment of pulmonary inflammation and influenza-associated pneumonia (21–25). *Haemophilus*, a predominant lower respiratory tract pathogen (26), is a leading cause of pediatric community-acquired pneumonia, capable of inducing localized suppurative infections or progressing to life-threatening systemic invasive diseases (27). Clinical manifestations include upper respiratory tract infections, urinary tract infections, and severe complications such as meningitis or sepsis (28). *Porphyromonas*, obligate anaerobic gram-negative bacilli, are well-documented contributors to periodontal infections, abscess formation, pelvic inflammatory disease, and antibiotic-resistant postoperative infections (29). Animal studies indicate *Prevotella* colonization exacerbates experimental colitis by upregulating pro-inflammatory cytokines (IL-6, TNF-α) and modulating anti-inflammatory IL-10 secretion, thereby highlighting its critical role in the pathogenesis of intestinal inflammation (30).

The enrichment of *Rothia* in MPA infections may serve not only as a biomarker but also as a synergistic factor in disease progression. Metagenomic evidence (31) reveals physical interactions between *Porphyromonas* FimA pili and *P. aeruginosa* alginate

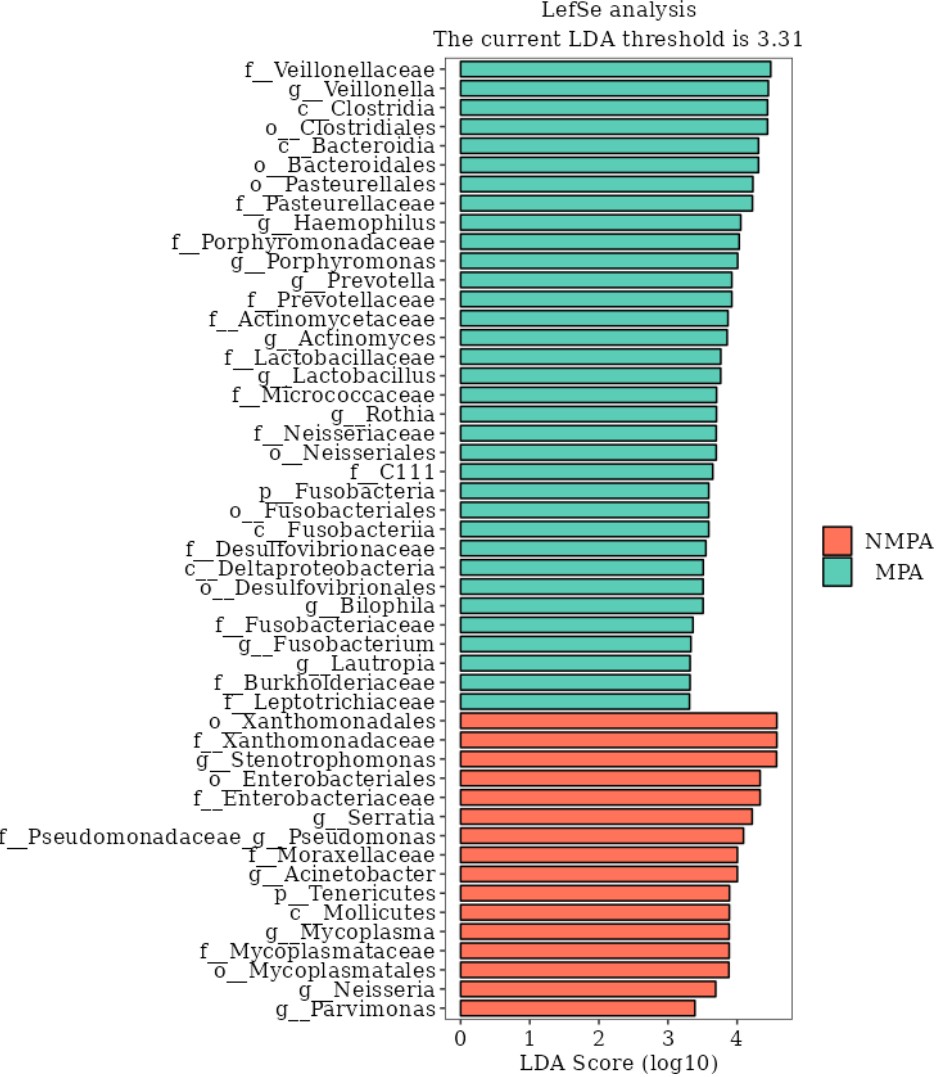

**FIG 4** LEfSe analysis diagram of microbiota in the NMPA and MPA groups. The LEfSe analysis shows the LDA score of the microbial taxonomic units in the NMPA (red column) and MPA (green column) sample groups. The higher the score (the longer the column), the more significant the enrichment degree of the taxonomic unit in the corresponding group.

matrices within polymicrobial biofilms, suggesting a mechanism underlying persistent co-infections.

Elevated serum levels of IL-6, IL-8, TNF-α, IL-10, and IL-17 were observed in the MPA group ($P < 0.05$), indicating that mucoid infections exacerbate pulmonary tissue injury through the amplification of inflammatory responses. This state of dysregulated immunomodulation may not only facilitate colonization by *Pseudomonas aeruginosa* but also propagate downstream inflammatory cascades, ultimately leading to alveolar epithelial barrier dysfunction and pathological remodeling (32).

TNF-α, a pivotal immunoregulatory factor, drives naïve T-cell differentiation toward the Th2 lineage in adaptive immunity. However, excessive activation of the Th2 pathway results in overexpression of IL-6 and IL-4, disrupting local microenvironmental homeostasis and contributing to progressive tissue damage (33, 34). IL-6 exhibits dual-phase biological effects during infection: early-phase secretion enhances pathogen suppression, whereas sustained overexpression paradoxically promotes *Pseudomonas aeruginosa* proliferation and colonization (35). Notably, biofilm-forming strains of *P. aeruginosa* induce significantly higher IL-6 production compared to planktonic isolates, underscoring the role of biofilm structures in driving immune dysregulation (36).

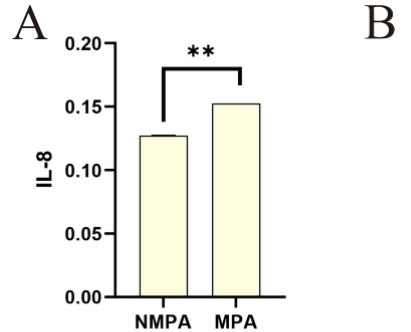
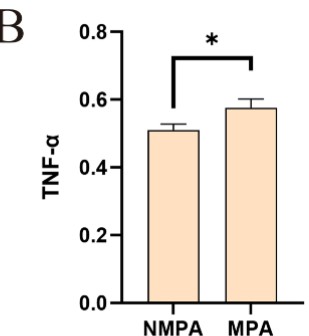
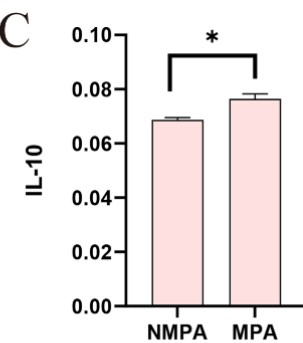
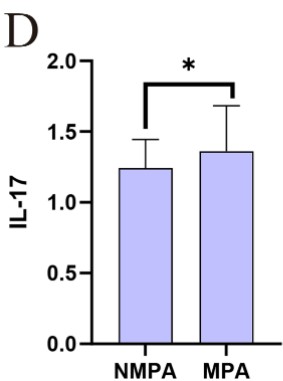
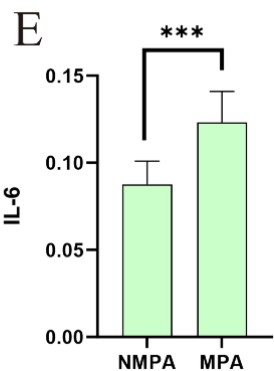

**FIG 5** Comparison of serum immune factor levels between the NMPA and MPA groups. Panels A to E represent the different inflammatory factors in sequence: IL-8 (A), TNF-α (B), IL-10 (C), IL-17 (D), and IL-6 (E). *$P < 0.05$, **$P < 0.01$, ***$P < 0.001$.

The mucoid phenotype is characterized by biofilm architectures enriched with pathogen-associated molecular patterns, including extracellular DNA, lipopolysaccharides, and polysaccharides. These components potently activate neutrophil and macrophage responses independent of viable bacteria, driving intense inflammation (37). MPA-induced lymphocyte overactivation is closely associated with pathological overexpression of IL-6, which may in turn promote bacterial persistence, thus creating a self-perpetuating cycle that contributes to the establishment and maintenance of chronic infection (38).

IL-10, a key anti-inflammatory mediator, modulates immune hyperactivation and supports mucosal barrier integrity via host-microbiota crosstalk (39). Conversely, IL-17 elevation driven by increased microbial load exacerbates pulmonary inflammation, as evidenced by IL-17 neutralization studies attenuating lung injury in murine models (40).

Our heatmap analysis demonstrated significant positive correlations between *Veillonella*, *Rothia*, *Porphyromonas*, and *Prevotella* and the proinflammatory cytokines IL-10, TNF-α, IL-17, and IL-6 in MPA-infected patients. *Veillonella,* a predominant commensal bacterium in the oral and gut microbiota, exhibits dual immunomodulatory properties. Mechanistically, it may indirectly activate TLR pathways (such as through co-aggregation with pathogens in periodontitis, driving IL-1β/IL-8 release [41]), while its short-chain fatty acids (SCFAs) promote Treg-mediated IL-10 secretion to attenuate excessive inflammatory responses (42). Paradoxically, elevated IL-10 may impair pathogen clearance, as observed in *Pseudomonas aeruginosa* infection models (43). Clinical studies have associated *Veillonella* enrichment in COPD airways with IL-10 dysregulation (44) and with TNF-α/IL-17 elevation via TLR2/4 activation and Th17 polarization, exacerbating lung injury (42, 45).

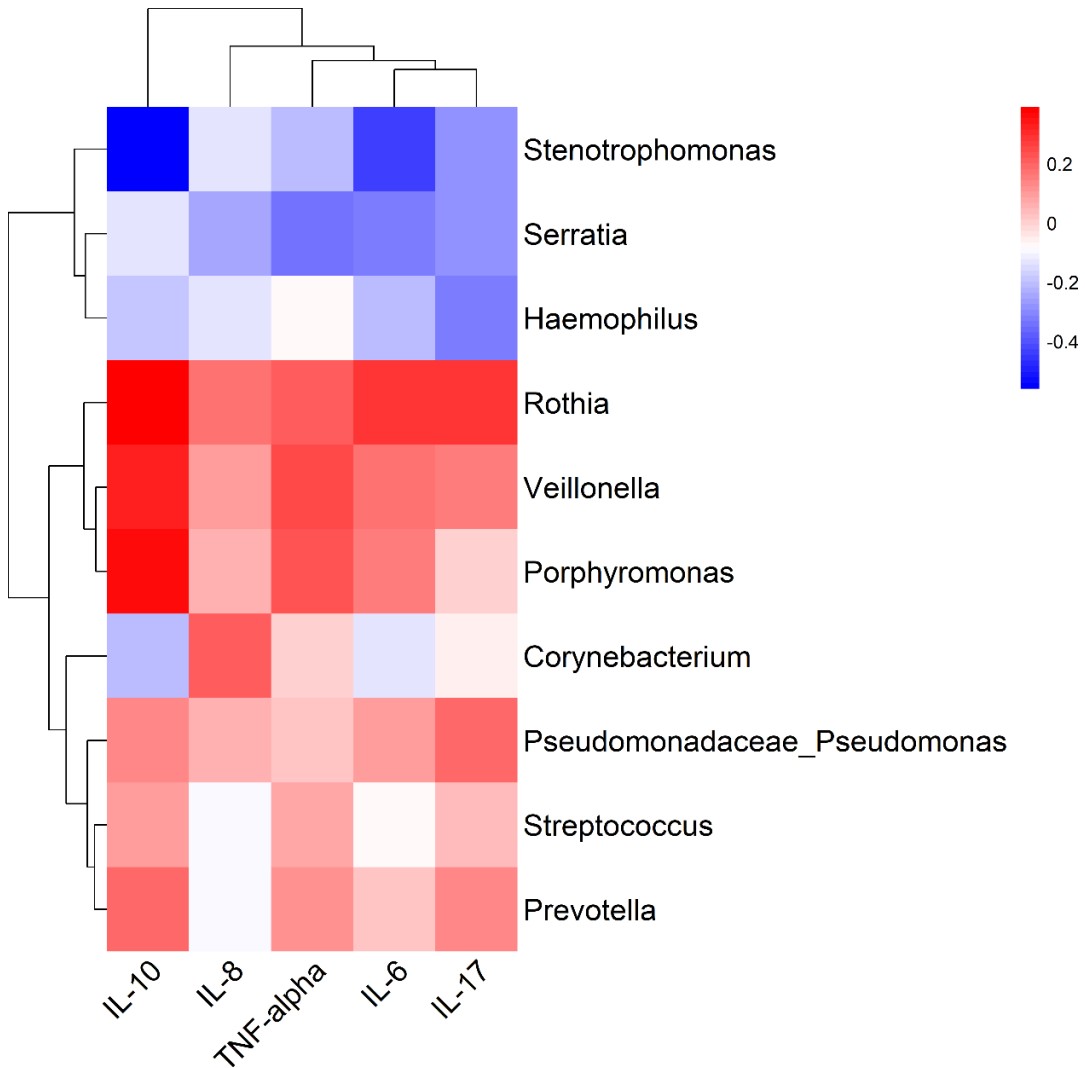

**FIG 6** Correlation between serum immune factor levels and the abundance of differential bacteria. The heatmap illustrates the correlation between the oral microbiota (shown in the right column) and inflammatory factors (listed in the bottom row), where positive correlations are represented in red, negative correlations in blue, and non-significant associations are approximated by white.

Notably, MPA and *Veillonella* may appear to exert synergistic immunomodulation in chronic infections: SCFAs from *Veillonella* promote an IL-10-mediated immune-evasive environment, while MPA-derived alginate amplifies TLR/NF-κB signaling, resulting in the coordinated upregulation of TNF-α, IL-6, and IL-17 and driving neutrophil-dominated inflammation (46). These findings are supported by clinical evidence linking the co-enrichment of MPA and *Veillonella* with increased disease severity (47).

While this study offered valuable insights, several limitations warrant acknowledgment. First, the single-center design may restrict the generalizability of the findings to other clinical settings. Second, the relatively modest sample size limited statistical power in detecting subtle microbial-immune interactions. Third, potential confounding factors were not fully accounted for. Furthermore, the case-control design did not permit causal inference regarding microbiota-immune associations. Future multicenter studies involving larger cohorts, longitudinal sampling, and standardized methodologies are required to confirm and extend these findings.

## Summary

In conclusion, the clinically distinct microbial ecosystems in mucoid PA correlate with exacerbated inflammation, while non-mucoid PA's microbiome associates with attenuated responses. This phenotype-driven dichotomy provides actionable biomarkers for stratified antimicrobial/immunomodulatory therapies in chronic lung disease.

## ACKNOWLEDGMENTS

We are grateful to all the people who collected samples at the First Affiliated Hospital, Hainan Medical University.

This project was supported by the National Natural Science Foundation of China (82160634) and (72361127506), the Education Department of Hainan Province, project number (Hnjg2025ZD-37).

Conceptualization: Hairong Huang and Wenjuan Liang; Methodology: Yuanhua Lin; writing: Yuanhua Lin; Writing: Chenyan Zhang and Zhengke Sun; Writing: Hairong Huang, Wenjuan Liang, and Rongguang Zhang. All authors have read and agreed to the published version of the manuscript.

## AUTHOR AFFILIATIONS

[1]School of Public Health, Key Laboratory of Tropical Translational Medicine of Ministry of Education, Hainan Medical University, Haikou, Hainan, China
[2]Department of Clinical Laboratory, Hainan Medical University, First Affiliated Hospital, Haikou, Hainan, China

## AUTHOR ORCIDs

Wenjuan Liang http://orcid.org/0000-0001-9936-045X

## FUNDING

| Funder | Grant(s) | Author(s) |
|---|---|---|
| National Natural Science Foundation of China | 82160634 | Wenjuan Liang |
| National Natural Science Foundation of China | 72361127506 | Rongguang Zhang |
| Education Department of Hainan Province | Hnjg2025ZD-37 | Wenjuan Liang |

## AUTHOR CONTRIBUTIONS

Chenyan Zhang, Data curation, Resources | Zhengke Sun, Methodology, Resources | Yuanhua Lin, Methodology, Resources | Wenfang Long, Formal analysis, Visualization | Rongguang Zhang, Formal analysis, Supervision, Writing – review and editing | Hairong Huang, Conceptualization, Supervision, Validation, Writing – review and editing | Wenjuan Liang, Funding acquisition, Supervision, Writing – original draft, Writing – review and editing

## DATA AVAILABILITY

All data contained in this study can be obtained from the website (https://www.ncbi.nlm.nih.gov/Traces/study/?acc=SRP628432) or the corresponding author (Wen Juan Liang, MD, E-mail: wenwen3_1@126.com) upon reasonable request.

## ETHICS APPROVAL

The study protocol was approved by the Ethics Committee of Hainan Medical University (HYLL-2024-059), and informed consent was obtained from all participants.

Qualified sputum samples (1 mL) were cryopreserved at −80℃ for subsequent 16S rDNA sequencing, in accordance with the protocols specified in the National Clinical Laboratory Standardization Procedures (4th Edition).

All the authors consented to its submission for publication.

## ADDITIONAL FILES

The following material is available online.

### Supplemental Material

**Supplemental material (Spectrum02295-25-S0001.pdf).** Table S1; Fig. S1 to S3.

### Open Peer Review

**PEER REVIEW HISTORY (review-history.pdf).** An accounting of the reviewer comments and feedback.

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
