## [Reviewer comments · Microbiology Spectrum]

Microbiology Spectrum

Pulmonary Muroid *Pseudomonas aeruginosa* Infection and Association with Higher species richness and stronger inflammatory immune response

Chenyan Zhang, Zhengke Sun, Yuanhua Lin, Wenfang Long, Rongguang Zhang, Hairong Huang, and Wenjuan Liang

Corresponding Author(s): Wenjuan Liang, Hainan Medical University

Review Timeline:

Submission Date:	August 2, 2025
Editorial Decision:	September 19, 2025
Revision Received:	January 12, 2026
Accepted:	January 24, 2026

Editor: Ping Ren

Reviewer(s): Disclosure of reviewer identity is with reference to reviewer comments included in decision letter(s). The following individuals involved in review of your submission have agreed to reveal their identity: Shuaiyin Chen (Reviewer #1); Yongbin Wang (Reviewer #2)

Transaction Report:

DOI: <https://doi.org/10.1128/spectrum.02295-25>

Re: Spectrum02295-25 (**Pulmonary Mucoïd Pseudomonas aeruginosa Infection and Association with Higher species richness and stronger inflammatory response immune**)

Dear Prof. Wenjuan Liang:

Thank you for the privilege of reviewing your work. Below you will find my comments, instructions from the Spectrum editorial office, and the reviewer comments.

Revision Guidelines

Sincerely,
Ping Ren
Editor
Microbiology Spectrum

Reviewer #1 (Comments for the Author):

The study provided evidence of clinically distinct microbial ecosystems in mucoïd P. aeruginosa correlated with exacerbated inflammation, while non-mucoïd PA's microbiome associated with attenuated responses. This phenotype-driven dichotomy may provide actionable biomarkers for stratified antimicrobial/immunomodulatory therapies in chronic lung disease.

Major revision:

Title:

1. The title is somewhat awkward and incomplete. The phrase "Stronger Inflammatory Response Immune" lacks a clear logical connection; it is recommended to revise it to "Stronger Immune Inflammatory Response" to make the expression more accurate and in line with academic writing norms.

Abstract

2. Species richness description: When mentioning "significantly higher α -diversity indices in species richness (Chao1 and Observed-species, $P < 0.05$)", it is advisable to briefly add the specific fold difference or numerical range of the Chao1 and Observed-species indices between the two groups.

3. The abstract states that the findings "provide actionable biomarkers for stratified antimicrobial/immunomodulatory therapies", but it does not specify which specific microbial taxa (e.g., *Veillonella* spp., *Rothia* spp.) or cytokine profiles (e.g., elevated TNF- α , IL-6) can be used as these biomarkers. Supplementing this information will enhance the practical guiding significance of the abstract.

Introduction

4. In the first paragraph, there is a lack of citations of high-impact studies on the mechanism of mucoid PA biofilm formation or the interaction between PA and the host immune system published in the past 2-3 years (2022-2024). Please add such literature will enrich the theoretical basis of the study and highlight the novelty of this research.

5. Although it is mentioned that "current PA research predominantly focuses on single phenotypes, lacking parallel comparisons between MPA and NMPA", the specific effects of this research gap on clinical practice or scientific research (e.g., inaccurate clinical treatment strategies for different PA phenotypes, unclear pathogenesis of MPA infections) are not elaborated.

Methods

6. Study population definition: The inclusion criterion is "hospitalized patients with PA pneumonia", but the diagnostic criteria for PA pneumonia (e.g., whether it is based on clinical symptoms, imaging findings, and microbiological detection results; specific microbiological detection methods such as sputum culture positive times) are not clearly stated. Defining this will ensure the consistency and reproducibility of the study population.

7. Sputum sample processing: When describing "qualified samples (1 mL) were preserved at -80°C ", when collect sputum sample? This information is crucial for the reliability of subsequent microbiota analysis results, so it should be supplemented.

8. Statistical analysis : The study adopts a case-control design with 40 cases in the NMPA group and 20 cases in the MPA group. However, there is no explanation of whether the sample size is determined based on sample size calculation.

Results

9. Demographic and clinical characteristics: Table S1 (Supplementary Material) is mentioned to show no significant differences in demographic and clinical characteristics between the two groups. Adding a concise summary of key data in the text will help readers quickly understand the baseline characteristics of the study population without relying on supplementary materials.

10. Please report the specific the P-values in results.

Discussion

11. Mechanism explanation depth: When discussing the correlation between specific microbial taxa and immune factors (e.g., *Veillonella* spp. positively correlated with IL-10/TNF- α /IL-17/IL-6), the specific molecular mechanisms underlying these correlations are not deeply explored.

12. Limitations: The study mentions several limitations, such as single-center design, small sample size, and cross-sectional design, but does not propose specific improvement measures for these limitations.

Figures and Tables

13. Figure legends: The legends of some figures (e.g., Figure 1, Figure 2) are too concise and do not clearly explain the meaning of each component in the figures. Please enrich the figure legends to ensure that readers can understand the content of the figures without relying on the text.

14. In Figure 5, which shows the differences in serum immune factor levels between the two groups, the error bars are not clearly labeled. Additionally, the specific numerical values of the immune factor levels in each group are not provided in the text.

Language

15. The language in the manuscript also needs polishing.

Reviewer #2 (Comments for the Author):

Comments to the manuscript: Pulmonary Mucoid *Pseudomonas aeruginosa* Infection and Association with Higher species richness and stronger inflammatory response immune

This study done by Liang et al. is interesting, which aims to address an important gap in understanding PA phenotype-specific microbiota-immune interactions, presenting a good idea. I think this manuscript is able to do better by doing some revisions.

Revision:

METHOD

1. Research design and Sample Size

The study included 20 mucoid *Pseudomonas aeruginosa* (MPA) patients and 40 non-mucoid (NMPA) patients, Whether the

sample size is sufficient and whether it is an ordinary case-control study or a matched case-control study.

2. Confounding Variables of Antibiotic Exposure

While the manuscript mentions matching for age, sex, and clinical characteristics, critical confounders like antibiotic exposure history is not addressed. Antibiotics can profoundly alter respiratory microbiota, and disease severity may independently influence inflammation.

3. Sputum Sample Quality

The manuscript stated that "qualified samples (1 mL)" were used but does not define qualification criteria, which are standard for ensuring lower respiratory tract origin.

DISCUSSION

4. Ambiguities in α -Diversity Interpretation

The MPA group exhibits higher Chao1 and Observed-species indices ($P < 0.05$) but no difference in Shannon diversity ($P = 0.41$). This discrepancy (richness vs. evenness) is not discussed. Higher richness with similar evenness could indicate colonization by rare taxa, but the biological relevance is unclear.

6. Mechanistic Weakness in Microbe-Cytokine Correlations

Spearman correlations show associations between taxa (e.g., *Veillonella*) and cytokines (e.g., IL-10), but no mechanistic explanation is provided.

7. FIGURE LEGEND

Please add the figure legend.

The study provided evidence of clinically distinct microbial ecosystems in mucoid *P. aeruginosa* correlated with exacerbated inflammation, while non-mucoid PA's microbiome associated with attenuated responses. This phenotype-driven dichotomy may provide actionable biomarkers for stratified antimicrobial/immunomodulatory therapies in chronic lung disease.

Major revision:

Title:

1. The title is somewhat awkward and incomplete. The phrase "Stronger Inflammatory Response Immune" lacks a clear logical connection; it is recommended to revise it to "Stronger Immune Inflammatory Response" to make the expression more accurate and in line with academic writing norms.

Abstract

2. Species richness description: When mentioning "significantly higher α -diversity indices in species richness (Chao1 and Observed-species, $P < 0.05$)", it is advisable to briefly add the specific fold difference or numerical range of the Chao1 and Observed-species indices between the two groups.

3. The abstract states that the findings "provide actionable biomarkers for stratified antimicrobial/immunomodulatory therapies", but it does not specify which specific microbial taxa (e.g., *Veillonella* spp., *Rothia* spp.) or cytokine profiles (e.g., elevated TNF- α , IL-6) can be used as these biomarkers. Supplementing this information will enhance the practical guiding significance of the abstract.

Introduction

4. In the first paragraph, there is a lack of citations of high-impact studies on the mechanism of mucoid PA biofilm formation or the interaction between PA and the host immune system published in the past 2-3 years (2022-2024). Please add such literature will enrich the theoretical basis of the study and highlight the novelty of this research.

5. Although it is mentioned that "current PA research predominantly focuses on

single phenotypes, lacking parallel comparisons between MPA and NMPA", the specific effects of this research gap on clinical practice or scientific research (e.g., inaccurate clinical treatment strategies for different PA phenotypes, unclear pathogenesis of MPA infections) are not elaborated.

Methods

6. Study population definition: The inclusion criterion is "hospitalized patients with PA pneumonia", but the diagnostic criteria for PA pneumonia (e.g., whether it is based on clinical symptoms, imaging findings, and microbiological detection results; specific microbiological detection methods such as sputum culture positive times) are not clearly stated. Defining this will ensure the consistency and reproducibility of the study population.

7. Sputum sample processing: When describing "qualified samples (1 mL) were preserved at -80°C ", when collect sputum sample? This information is crucial for the reliability of subsequent microbiota analysis results, so it should be supplemented.

8. Statistical analysis : The study adopts a case-control design with 40 cases in the NMPA group and 20 cases in the MPA group. However, there is no explanation of whether the sample size is determined based on sample size calculation.

Results

9. Demographic and clinical characteristics: Table S1 (Supplementary Material) is mentioned to show no significant differences in demographic and clinical characteristics between the two groups. Adding a concise summary of key data in the text will help readers quickly understand the baseline characteristics of the study population without relying on supplementary materials.

10. Please report the specific the P-values in results.

Discussion

11. Mechanism explanation depth: When discussing the correlation between specific microbial taxa and immune factors (e.g., *Veillonella* spp. positively correlated with IL-10/TNF- α /IL-17/IL-6), the specific molecular mechanisms underlying these correlations are not deeply explored.

12. Limitations: The study mentions several limitations, such as single-center design,

small sample size, and cross-sectional design, but does not propose specific improvement measures for these limitations.

Figures and Tables

13. Figure legends: The legends of some figures (e.g., Figure 1, Figure 2) are too concise and do not clearly explain the meaning of each component in the figures. Please enrich the figure legends to ensure that readers can understand the content of the figures without relying on the text.

14. In Figure 5, which shows the differences in serum immune factor levels between the two groups, the error bars are not clearly labeled. Additionally, the specific numerical values of the immune factor levels in each group are not provided in the text.

Language

15. The language in the manuscript also needs polishing.

Respondent Letter

Dear Editor and Reviewers,

We sincerely appreciate the time and effort you have dedicated to reviewing our manuscript titled "Pulmonary Muroid *Pseudomonas aeruginosa* Infection and Association with Higher species richness and stronger inflammatory response immune". We are grateful for your constructive comments and valuable suggestions, which have helped us significantly improve the quality of our work. Below, we provide a point-by-point response to all the comments and revision guidelines raised by the reviewers. All modifications made to the manuscript have been highlighted in the revised version for your convenience.

Sincerely yours,

Wenjuan Liang, PhD, Prof.

October 22, 2025

Reviewer #1 (Comments for the Author):

The study provided evidence of clinically distinct microbial ecosystems in mucoid *P. aeruginosa* correlated with exacerbated inflammation, while non-mucoid PA's microbiome associated with attenuated responses. This phenotype-driven dichotomy may provide actionable biomarkers for stratified antimicrobial/immunomodulatory therapies in chronic lung disease.

Major revision:

Title:

1. The title is somewhat awkward and incomplete. The phrase "Stronger Inflammatory Response Immune" lacks a clear logical connection; it is recommended to revise it to "Stronger Immune Inflammatory Response" to make the expression more accurate and in line with academic writing norms.

Response: Many thanks for your reminder. We have revised the title “ Pulmonary Mucoid *Pseudomonas aeruginosa* Infection and Association with Higher species richness and stronger inflammatory immune response ”.

Abstract

2. Species richness description: When mentioning "significantly higher α -diversity indices in species richness (Chao1 and Observed-species, $P < 0.05$)", it is advisable to briefly add the specific fold difference or numerical range of the Chao1 and Observed-species indices between the two groups.

Response: Thank you for your suggestion. We added the revised as below:

Compared to the non-mucoid PA group, the mucoid PA group demonstrated significantly higher α -diversity indices in terms of species richness, as indicated by the Chao1 ($P = 0.0015$) and Observed-species metrics ($P = 0.0014$, respectively). Furthermore, distinct β -diversity patterns were observed between the two groups ($P < 0.05$).

The mucoid PA infections showed marked elevation of IL-8 ($P = 0.0137$), TNF- α ($P = 0.0048$), IL-10 ($P = 0.0042$), IL-17 ($P = 0.0220$), and IL-6 ($P = 0.0001$).

3. The abstract states that the findings "provide actionable biomarkers for stratified antimicrobial/immunomodulatory therapies", but it does not specify which specific microbial taxa (e.g., *Veillonella* spp., *Rothia* spp.) or cytokine profiles (e.g., elevated TNF- α , IL-6) can be used as these biomarkers. Supplementing this information will enhance the practical

guiding significance of the abstract.

Response: We sincerely appreciated the reviewer's insightful suggestion regarding the need to specify actionable biomarkers for stratified therapies.

The differences in microbial taxa and cytokine profiles between the non-mucoid *Pseudomonas aeruginosa* (PA) group and the mucoid PA group have been thoroughly described in our results; therefore, key microbial taxa (e.g., *Veillonella* spp., *Rothia* spp.) and cytokine profiles (e.g., elevated TNF- α , IL-6) are not reiterated in the abstract conclusion.

Introduction

4. In the first paragraph, there is a lack of citations of high-impact studies on the mechanism of mucoid PA biofilm formation or the interaction between PA and the host immune system published in the past 2-3 years (2022-2024). Please add such literature will enrich the theoretical basis of the study and highlight the novelty of this research.

Response: We sincerely appreciated the reviewer's insightful suggestion regarding the need to incorporate recent high-impact studies on mucoid PA biofilm formation and host immune interactions.

Our study aimed to contextualize the clinical significance of mucoid PA infections, particularly emphasizing their role in persistent airway colonization and therapeutic failure in chronic lung disease. In addition,

the study uniquely focused on the comparative analysis of mucoid (MPA) vs. non-mucoid (NMPA) PA in microbiota-immune interactions, an aspect not extensively explored in recent literature. We understood the reviewer's intent to enrich the abstract's academic rigor, and we assured that the study's design and results were rigorously supported by existing literature. We hoped this explanation clarifies our position and appreciated the opportunity to discuss this further.

5. Although it is mentioned that "current PA research predominantly focuses on single phenotypes, lacking parallel comparisons between MPA and NMPA", the specific effects of this research gap on clinical practice or scientific research (e.g., inaccurate clinical treatment strategies for different PA phenotypes, unclear pathogenesis of MPA infections) are not elaborated.

Response: We sincerely thank the reviewer for this insightful comment regarding the need to elaborate on the specific implications of the research gap. To address this concern, we have added the following elaboration in the manuscript:

"The prevailing focus on single-phenotype PA models has obscured critical differences in pathogenesis and therapeutic susceptibility between MPA and NMPA. Clinically, this gap contributes to undifferentiated treatment protocols and persistent infections in chronic respiratory

diseases. "

We hope these clarifications adequately address the reviewer's concern.

Methods

6. Study population definition: The inclusion criterion is "hospitalized patients with PA pneumonia", but the diagnostic criteria for PA pneumonia (e.g., whether it is based on clinical symptoms, imaging findings, and microbiological detection results; specific microbiological detection methods such as sputum culture positive times) are not clearly stated. Defining this will ensure the consistency and reproducibility of the study population.

Response: Thank you for your suggestion. Our diagnostic criteria included the clinical symptoms, imaging findings and microbiological detection results. For a clearer expression, we revised as below:

Hospitalized patients with pneumonia, confirmed by positive *P.aeruginosa* isolation in two consecutive cultures and characteristic imaging findings, admitted from August 2023 to January 2024 at a tertiary hospital in Haikou city, were enrolled, according to Guidelines for the Diagnosis and Treatment of Community-Acquired Pneumonia in Chinese Adults (2016 Edition).

7. Sputum sample processing: When describing "qualified samples (1 mL)

were preserved at -80°C , when collect sputum sample? This information is crucial for the reliability of subsequent microbiota analysis results, so it should be supplemented.

Response: Thank you for your suggestion. We revised as below:

Deep sputum samples from the lower respiratory tract, collected after oral decontamination, were obtained in the morning and cultured on MacConkey (MAC) agar at 35°C for 24 hours to identify dominant pathogens. Qualified sputum samples (1 mL) were cryopreserved at -80°C for subsequent 16S rDNA sequencing, in accordance with the protocols specified in the National Clinical Laboratory Standardization Procedures (4th Edition).

The qualified sputum samples from the lower respiratory tract, collected after oral decontamination, were obtained in the morning.

8. Statistical analysis : The study adopts a case-control design with 40 cases in the NMPA group and 20 cases in the MPA group. However, there is no explanation of whether the sample size is determined based on sample size calculation.

Response: Thank you for your suggestion.

The study enrolled 20 mucoid *Pseudomonas aeruginosa* (MPA) patients and 40 non-mucoid (NMPA) patients. Regarding the adequacy of this sample size:

(1) Challenges in Microbial Sequencing Studies: Traditional sample size calculations (e.g., power analysis based on core parameters like effect size) are often impractical for microbiome research due to: High variability in microbial community structures and multidimensional data (α/β -diversity, differential taxa).

(2) Rationale for the Chosen Sample Size:

① Precedent from Similar Studies: The sample size aligns with prior microbiome studies comparing phenotypic variants in infections (e.g., COVID-19 or TB)^[1-2], where 5-15 samples per group are common for detecting significant diversity differences.

② Statistical Significance Achieved: Despite the modest size, the study identified significant differences in α -diversity (Chao1, $P < 0.05$), β -diversity (PERMANOVA, $P < 0.05$) and Cytokine levels (TNF- α , IL-6, etc., $P < 0.05$).

[1] Pérez-Nicado R, Massa C, Rodríguez-Noda LM, Müller A, Puga-Gómez R, Ricardo-Delgado Y, Paredes-Moreno B, Rodríguez-González M, García-Ferrer M, Palmero-Álvarez I, Garcés-Hechavarría A, Rivera DG, Valdés-Balbín Y, Vérez-Bencomo V, García-Rivera D, Seliger B. Comparative Immune Response after Vaccination with SOBERANA® 02 and SOBERANA® plus Heterologous Scheme and Natural Infection in Young Children. *Vaccines (Basel)*. 2023 Oct 25;11(11):1636. doi: 10.3390/vaccines11111636.

[2] Yang X, Yan J, Xue Y, Sun Q, Zhang Y, Guo R, Wang C, Li X, Liang Q, Wu H, Wang C, Liao X, Long S, Zheng M, Wei R, Zhang H, Liu Y, Che N, Luu LDW, Pan J, Wang G, Wang Y. Single-cell profiling reveals distinct immune response landscapes in tuberculous pleural effusion and non-TPE. *Front Immunol*. 2023 Jun 26;14:1191357. doi: 10.3389/fimmu.2023.1191357.

Results

9. Demographic and clinical characteristics: Table S1 (Supplementary

Material) is mentioned to show no significant differences in demographic and clinical characteristics between the two groups. Adding a concise summary of key data in the text will help readers quickly understand the baseline characteristics of the study population without relying on supplementary materials.

Response: Thank you for your suggestion. We revised as below:

A total of 60 patients were enrolled in the study, comprising 40 in the NMPA group and 20 in the MPA group. As detailed in Table S1 (Supplementary Material), no statistically significant differences were observed between the two groups in baseline characteristics, including demographics (sex, female: 70.0% vs 50.3%; age: 65.55±14.17 vs 66.78±14.81 years, smoking, alcohol use), comorbidities (hypertension, diabetes mellitus, cerebrovascular disease, or cardiac disorders), or hematologic parameters ($P > 0.05$ for all group comparisons).

10. Please report the specific the P-values in results.

Response: Thank you for your suggestion. We revised as below:

Compared to the non-mucoid PA group, the mucoid PA group demonstrated significantly higher α -diversity indices in terms of species richness, as indicated by the Chao1 (284.64±158.70 vs 175.16±143.85, $P = 0.0015$) and Observed-species metrics (265.64±149.81 vs 164.51±140.93, $P = 0.0014$)

Discussion

11.Mechanism explanation depth: When discussing the correlation between specific microbial taxa and immune factors (e.g., *Veillonella* spp. positively correlated with IL-10/TNF- α /IL-17/IL-6), the specific molecular mechanisms underlying these correlations are not deeply explored.

Response: We sincerely thank the reviewer for this insightful comment.

In the original manuscript's Discussion section, we had already addressed potential mechanisms through the following points: Activation of Toll-like Receptor (TLR) Pathways, Metabolite-Mediated Immune Regulation and Biofilm-Mediated Synergistic Enhancement Mechanism. More specific comments are in-line below.

Our heatmap analysis demonstrated significant positive correlations between *Veillonella*, *Rothia*, *Porphyromonas*, and *Prevotella* and the proinflammatory cytokines IL-10, TNF- α , IL-17, and IL-6 in MPA-infected patients. *Veillonella*, a predominant commensal bacterium in the oral and gut microbiota, exhibits dual immunomodulatory properties. Mechanistically, it may indirectly activate TLR pathways (such as through via co-aggregation with pathogens in periodontitis, driving IL-1 β /IL-8 release [41]), while its short-chain fatty acids (SCFAs) promote Treg-mediated IL-10 secretion to attenuating excessive

inflammatory responses [45]. Paradoxically, elevated IL-10 may impair pathogen clearance, as observed in *Pseudomonas aeruginosa* infection models [42]. Clinical studies have associated with *Veillonella* enrichment in COPD airways with IL-10 dysregulation [43] and to TNF- α /IL-17 elevation via TLR2/4 activation and Th17 polarization, exacerbating lung injury [44,45].

Notably, MPA and *Veillonella* may appear to exert synergistic immunomodulation in chronic infections: SCFAs from *Veillonella* promote an IL-10-mediated immune-evasive environment, while MPA-derived alginate amplifies TLR/NF- κ B signaling, resulting in the coordinated upregulation of TNF- α , IL-6, and IL-17 and driving neutrophil-dominated inflammation[46]. These findings are supported by clinical evidence linking the co-enrichment of MPA and *Veillonella* with increased disease severity [47].

12.Limitations: The study mentions several limitations, such as single-center design, small sample size, and cross-sectional design, but does not propose specific improvement measures for these limitations.

Response: Thank you for your suggestion. We were aware that our research had certain limitations, so we also proposed the directions that future research should pay attention to, such as future multicenter studies involving larger cohorts, longitudinal sampling, and standardized methodologies are required to confirm and extend these findings.

Figures and Tables

13. Figure legends: The legends of some figures (e.g., Figure 1, Figure 2) are too concise and do not clearly explain the meaning of each component in the figures. Please enrich the figure legends to ensure that readers can understand the content of the figures without relying on the text.

Response: Thank you for your suggestion. We revised as below:

We have added the figure legend in Figures such as below:

Figure 1 α and β diversity of the sputum microbiota in the MPA and the NMPA group

Figure 1A displayed the distribution differences of Chao1, Good's Coverage, Shannon, and Observed-species, across the MPA and the NMPA group with boxplots. Figure 1B showed a distribution pattern of partial overlap but overall separation trend, indicating differences in microbial community structure between the two groups.

Figure 2 Composition of the microbiota at the genus level in the NMPA and MPA group

The stacked bar chart visually displayed the relative abundance (%) of bacterial phyla in the MPA and the NMPA group, and the legend on the right clearly labels the colors corresponding to each bacterial phylum, as the *Proteobacteria* is the dominant group and the second most abundant

phylum is *Firmicutes*.

Figure 3 Composition of the microbiota at the species level in the NMPA and MPA group

The relative abundance (%) of the genus level in the MPA and the NMPA group was displayed, and the top taxa in the NMPA group was *Corynebacterium*, whereas the MPA group exhibited dominance of *Veillonella*.

Figure 4 Lefse analysis diagram of microbiota in the NMPA and MPA group

The Lefse analysis showed the LDA score of the microbial taxonomic unit in the NMPA (red column) and MPA (green column) sample groups. The higher the score (the longer the column), the more significant the enrichment degree of the taxonomic unit in the corresponding group.

Figure 5 Comparison of serum immune factor levels between NMPA and MPA group. A to E represented the different inflammatory factors in sequence: IL-8 (A), TNF- α (B), IL-10 (C), IL-17 (D), and IL-6 (E).

* $P < 0.05$, ** $P < 0.01$, *** $P < 0.001$

Figure 5 Comparison of serum immune factor levels between NMPA and MPA group. A to E represented the different inflammatory factors in

sequence: IL-8 (A), TNF- α (B), IL-10 (C), IL-17 (D), and IL-6 (E).

* $P < 0.05$, ** $P < 0.01$, *** $P < 0.001$

Figure 6 Correlation between serum immune factor levels and the abundance of differential bacteria.

The heatmap illustrated the correlation between the oral microbiota (shown in the right column) and inflammatory factors (listed in the bottom row), where positive correlations were represented in red, negative correlations in blue, and non-significant associations approximated by white.

14. In Figure 5, which shows the differences in serum immune factor levels between the two groups, the error bars are not clearly labeled. Additionally, the specific numerical values of the immune factor levels in each group are not provided in the text.

Response: Thank you for your suggestion. We added the error bars, and the specific numerical values of the immune factor levels in the text.

ELISA revealed significantly elevated serum levels of IL-8 (1.20 vs 0.94, $P=0.0137$), TNF- α (0.52 vs 0.50, $P=0.0048$), IL-10 (0.69 vs 0.65, $P=0.0042$), IL-17 (0.64 vs 0.61, $P=0.0220$), and IL-6 (1.20 vs 0.94, $P=0.0001$) in the MPA group compared to NMPA (Figure 5).

Language

15. The language in the manuscript also needs polishing.

Response: Thank you for your suggestion.

We have further polished the language to express more accurately in the revised manuscript. In addition, we have consulted a native English expert for proofreading to polished the language in the manuscript.

Reviewer #2 (Comments for the Author):

Comments to the manuscript: Pulmonary Mucoid Pseudomonas aeruginosa Infection and Association with Higher species richness and stronger inflammatory response immune

This study done by Liang et al. is interesting, which aims to address an important gap in understanding PA phenotype-specific microbiota-immune interactions, presenting a good idea. I think this manuscript is able to do better by doing some revisions.

Revision:

METHOD

1. Research design and Sample Size

The study included 20 mucoid Pseudomonas aeruginosa (MPA) patients

and 40 non-mucoid (NMPA)) patients, Whether the sample size is sufficient and whether it is an ordinary case-control study or a matched case-control study.

Response: Many thanks for your reminder.

(1) Study Design

We have checked and found the description about the study design was ambiguous, so we translated “A case-control design was adopted, categorizing participants into MPA group and NMPA group, with age, sex, and clinical characteristics matched between cohorts.” into “ This study employed a case-control design, in which participants were classified into the MPA group and the NMPA group. No statistically significant differences were observed in age, sex, or clinical characteristics between the two groups, indicating adequate comparability. ”

(2) Sample Size

The study enrolled 20 mucoid *Pseudomonas aeruginosa* (MPA) patients and 40 non-mucoid (NMPA) patients. Regarding the adequacy of this sample size:

(1) Challenges in Microbial Sequencing Studies: Traditional sample size calculations (e.g., power analysis based on core parameters like effect size) are often impractical for microbiome research due to: High variability in microbial community structures and multidimensional data (α/β -diversity, differential taxa).

(2) Rationale for the Chosen Sample Size:

③ Precedent from Similar Studies: The sample size aligns with prior microbiome studies comparing phenotypic variants in infections (e.g., COVID-19 or TB)^[1-2], where 5 – 15 samples per group are common for detecting significant diversity differences.

④ Statistical Significance Achieved: Despite the modest size, the study identified significant differences in α -diversity (Chao1, $P < 0.05$), β -diversity (PERMANOVA, $P < 0.05$) and Cytokine levels (TNF- α , IL-6, etc., $P < 0.05$).

[1] Pérez-Nicado R, Massa C, Rodríguez-Noda LM, Müller A, Puga-Gómez R, Ricardo-Delgado Y, Paredes-Moreno B, Rodríguez-González M, García-Ferrer M, Palmero-Álvarez I, Garcés-Hechavarría A, Rivera DG, Valdés-Balbín Y, Vérez-Bencomo V, García-Rivera D, Seliger B. Comparative Immune Response after Vaccination with SOBERANA® 02 and SOBERANA® plus Heterologous Scheme and Natural Infection in Young Children. *Vaccines (Basel)*. 2023 Oct 25;11(11):1636. doi: 10.3390/vaccines11111636.

[2] Yang X, Yan J, Xue Y, Sun Q, Zhang Y, Guo R, Wang C, Li X, Liang Q, Wu H, Wang C, Liao X, Long S, Zheng M, Wei R, Zhang H, Liu Y, Che N, Luu LDW, Pan J, Wang G, Wang Y. Single-cell profiling reveals distinct immune response landscapes in tuberculous pleural effusion and non-TPE. *Front Immunol*. 2023 Jun 26;14:1191357. doi: 10.3389/fimmu.2023.1191357.

2. Confounding Variables of Antibiotic Exposure

While the manuscript mentions matching for age, sex, and clinical characteristics, critical confounders like antibiotic exposure history is not addressed. Antibiotics can profoundly alter respiratory microbiota, and disease severity may independently influence inflammation.

Response: Many thanks for your reminder.

We appreciated the reviewer's astute observation regarding antibiotic exposure as a critical confounder. To address this concern: All respiratory samples were collected on the morning of hospital admission Day 1 before routine antibiotic administration.

3. Sputum Sample Quality

The manuscript stated that "qualified samples (1 mL)" were used but does not define qualification criteria, which are standard for ensuring lower respiratory tract origin.

Response: Many thanks for your reminder.

The qualified sputum samples from the lower respiratory tract, collected after oral decontamination, were obtained in the morning.

DISCUSSION

4. Ambiguities in α -Diversity Interpretation

The MPA group exhibits higher Chao1 and Observed-species indices ($P < 0.05$) but no difference in Shannon diversity ($P = 0.41$). This discrepancy (richness vs. evenness) is not discussed. Higher richness with similar evenness could indicate colonization by rare taxa, but the biological relevance is unclear.

Response: We appreciated the reviewer's insightful observation regarding

the divergence between richness (Chao1/Observed-species) and evenness (Shannon) indices in our MPA group analysis.

This study revealed that sputum microbiota in patients with MPA pulmonary infections exhibited higher α -diversity (species richness) compared to those with NMPA infections group, suggesting that MPA-infected airways develop distinct microbial community structures characterized by increased species richness without substantial changes in community evenness.

6. Mechanistic Weakness in Microbe-Cytokine Correlations

Spearman correlations show associations between taxa (e.g., *Veillonella*) and cytokines (e.g., IL-10), but no mechanistic explanation is provided.

Response: Many thanks for your reminder. In the original manuscript's Discussion section, we had already addressed potential mechanisms through the following points: Activation of Toll-like Receptor (TLR) Pathways, Metabolite-Mediated Immune Regulation and Biofilm-Mediated Synergistic Enhancement Mechanism. More specific comments are in-line below.

Our heatmap analysis demonstrated significant positive correlations between *Veillonella*, *Rothia*, *Porphyromonas*, and *Prevotella* and the proinflammatory cytokines IL-10, TNF- α , IL-17, and IL-6 in MPA-infected patients. *Veillonella*, a predominant commensal bacterium

in the oral and gut microbiota, exhibits dual immunomodulatory properties. Mechanistically, it may indirectly activate TLR pathways (such as through via co-aggregation with pathogens in periodontitis, driving IL-1 β /IL-8 release [41]), while its short-chain fatty acids (SCFAs) promote Treg-mediated IL-10 secretion to attenuating excessive inflammatory responses [45]. Paradoxically, elevated IL-10 may impair pathogen clearance, as observed in *Pseudomonas aeruginosa* infection models [42]. Clinical studies have associated with *Veillonella* enrichment in COPD airways with IL-10 dysregulation [43] and to TNF- α /IL-17 elevation via TLR2/4 activation and Th17 polarization, exacerbating lung injury [44,45].

Notably, MPA and *Veillonella* may appear to exert synergistic immunomodulation in chronic infections: SCFAs from *Veillonella* promote an IL-10-mediated immune-evasive environment, while MPA-derived alginate amplifies TLR/NF- κ B signaling, resulting in the coordinated upregulation of TNF- α , IL-6, and IL-17 and driving neutrophil-dominated inflammation[46]. These findings are supported by clinical evidence linking the co-enrichment of MPA and *Veillonella* with increased disease severity [47].

7. FIGURE LEGEND

Please add the figure legend.

Response: Many thanks for your reminder. Please refer to the 13 in the

reply to Reviewer 1.

Re: Spectrum02295-25R1 (**Pulmonary Mucoïd Pseudomonas aeruginosa Infection and Association with Higher species richness and stronger inflammatory response immune**)

Dear Prof. Wenjuan Liang:

Your manuscript has been accepted, and I am forwarding it to the ASM production staff for publication. Your paper will first be checked to make sure all elements meet the technical requirements. ASM staff will contact you if anything needs to be revised before copyediting and production can begin. Otherwise, you will be notified when your proofs are ready to be viewed.

Sincerely,
Ping Ren
Editor
Microbiology Spectrum

Reviewer #1 (Comments for the Author):

The author has made revisions to the reviewers' comments.